# Temporal Series Analysis of Population Cycle Threshold Counts as a Predictor of Surge in Cases and Hospitalizations during the SARS-CoV-2 Pandemic

**DOI:** 10.3390/v15020421

**Published:** 2023-02-02

**Authors:** Fernando Cava, Jesús San Román, Pablo Barreiro, Francisco Javier Candel, Francisco Javier Álvarez-Timón, David Melero, Nerea Coya, Raquel Guillén, David Cantarero-Prieto, Javier Lera-Torres, Noelia Cobo-Ortiz, Jesús Canora, Francisco Javier Martínez-Peromingo, Raquel Barba, María del Mar Carretero, Juan Emilio Losa, Antonio Zapatero

**Affiliations:** 1Department of Clinical Chemistry, Synlab, 28036 Madrid, Spain; 2Clinical Laboratory, Infanta Sofia University Hospital, UR Health, 28703 Madrid, Spain; 3Department of Medical Specialties and Public Health, Universidad Rey Juan Carlos, 28922 Madrid, Spain; 4Unit of Microbiology, Regional Laboratory of Public Health, Hospital Enfermera Isabel Zendal, 28055 Madrid, Spain; 5Clinical Microbiology and Infectious Diseases, IdISSC Health Institute, Hospital Universitario Clínico San Carlos, 28040 Madrid, Spain; 6Health Economics Research Group, University of Cantabria and IDIVAL, 39005 Santander, Spain; 7Department of Internal Medicine, Hospital Universitario de Fuenlabrada, Universidad Rey Juan Carlos, 28922 Madrid, Spain; 8Department of Geriatrics, Hospital Universitario Rey Juan Carlos, Universidad Rey Juan Carlos, 28922 Madrid, Spain; 9Department of Internal Medicine, Hospital Universitario Rey Juan Carlos, Universidad Rey Juan Carlos, 28922 Madrid, Spain; 10Infectious Diseases Unit, University Hospital Fundación Alcorcón, 28922 Madrid, Spain

**Keywords:** COVID-19, SARS-CoV-2, hospitalizations, cycle threshold, RT-PCR, pandemic

## Abstract

Tools to predict surges in cases and hospitalizations during the COVID-19 pandemic may help guide public health decisions. Low cycle threshold (CT) counts may indicate greater SARS-CoV-2 concentrations in the respiratory tract, and thereby may be used as a surrogate marker of enhanced viral transmission. Several population studies have found an association between the oscillations in the mean CT over time and the evolution of the pandemic. For the first time, we applied temporal series analysis (Granger-type causality) to validate the CT counts as an epidemiological marker of forthcoming pandemic waves using samples and analyzing cases and hospital admissions during the third pandemic wave (October 2020 to May 2021) in Madrid. A total of 22,906 SARS-CoV-2 RT-PCR-positive nasopharyngeal swabs were evaluated; the mean CT value was 27.4 (SD: 2.1) (22.2% below 20 cycles). During this period, 422,110 cases and 36,727 hospital admissions were also recorded. A temporal association was found between the CT counts and the cases of COVID-19 with a lag of 9–10 days (*p* ≤ 0.01) and hospital admissions by COVID-19 (*p* < 0.04) with a lag of 2–6 days. According to a validated method to prove associations between variables that change over time, the short-term evolution of average CT counts in the population may forecast the evolution of the COVID-19 pandemic.

## 1. Introduction

Three years after the first infections were identified in Wuhan (China), the severe acute respiratory coronavirus 2 (SARS-CoV-2) has caused globally more than 650 million confirmed cases and over seven million deaths [1]. Throughout the pandemic, the reverse transcription polymerase chain reaction (RT-PCR) has been the most sensitive and specific method for the surveillance and diagnosis of SARS-CoV-2 infection, although other methods, such as lateral flow antigen tests, may have acceptable accuracy for the detection of infectious cases, with shorter turn-around times as compared with a RT-PCR [2].

The cycle threshold (CT) value represents the number of RT-PCR cycles required to detect a positive signal. In RT-PCR-positive tests, the lower CT among the amplified genes has been proposed as a surrogate marker of the patient’s infectivity, since it is inversely related to their viral load in respiratory secretions; as such, increases of 3.3 points in the CT counts would correlate with an approximately 10-fold decrease in the amount of viral genetic material present in the sample [3]. It must be acknowledged that this estimation of RNA concentration from CT values is subject to certain variability factors, such as (i) the type and quality of the sample obtained, (ii) the media and reagents used for the transport and processing of samples, (iii) the presence of inhibitors, (iii) any delay in processing, (iv) the genetic targets used for the amplification of the RNA, and (v) the SARS-CoV-2 variants detected [4,5].

Still, these caveats mostly apply to the fact that a high CT value is not always indicative of a low risk of infectivity; a high CT value may also reflect that the patient is in the first days of infection, and therefore in the phase of rising viral RNA concentrations. Although RT-PCR does not detect viable viruses, low CT values correlate better with higher infectivity [6]; according to previous studies, SARS-CoV-2 could be cultured in more than 70% of the samples with a CT value below 25, but only from less than 3–8% of the samples with values equal to or above 35 [4,7]. Low CT counts may also provide prognostic information, as higher viral loads are related to more severe coronavirus infectious disease (COVID-19) [8,9] and to death after hospitalization [10].

From the epidemiological point of view, the analysis of CT counts in a population can provide a useful measure of the dynamics of COVID-19 at a given time; rising numbers of individuals found that low CT values could reflect the immediate growth of the epidemic in a community. Surveillance and simulation analyses have suggested that a reduction in the median value of CT precedes a local increase in the transmission of SARS-CoV-2 infections [11]. The median CT count may also be just as useful in predicting the severity of cases in populations as it is in predicting among individuals. It is also to be determined whether massive vaccination campaigns and changes in viral variants may influence the significance of the CT values. The objective of this study is to correlate the median value of the CT in the daily positive PCR nasopharyngeal samples with the evolution of the number of cases and the incidence of hospitalizations in the Community of Madrid during the third pandemic wave. We used the analysis of temporal series to analyze the connection between the curves of the CT counts and the curves of the COVD cases and hospitalizations, a type of test that is optimal to find associations between parameters that have a daily variation over a short period of time.

## 2. Methodology

### 2.1. Setting

The Community of Madrid is a region located in the center of Spain; with an area of 8028 km^2^, it represents the third most populated region in Spain, with more than 6.5 million inhabitants. It is assisted by 40 hospitals, 8 of which are considered to be highly complex. Primary care assistance is organized into 286 basic health zones (BHZs), each covering around 25,000 inhabitants served at 425 health care centers [12].

### 2.2. Period

Nasopharyngeal samples in subjects with suspected COVID-19, either considered as cases due to the presence of symptoms or contact with cases, from the primary care centers of three primary care directorates (105 centers), and the emergency services or outpatient clinics from six public hospitals in the metropolitan area of the city of Madrid were sent to the Central Reference Laboratory from the start of the pandemic to November 2021. Samples from hospitalized patients were not evaluated in this study, given that many of these PCR tests were carried out for follow-up or screening before elective procedures. Only positive SARS-CoV-2 RT-PCR samples were considered for this analysis.

The study covered the third pandemic wave, between 11 October 2020 and 2 May 2021, a period when the vaccination campaign was just starting, and the circulating variants were B.1.177 and B.1.1.7 (Alpha) [13].

Along with the SARS-CoV-2 RT-PCR qualitative results and the CT counts for each sample, the demographic data for tested individuals were age, sex, date of birth, and clinical setting.

### 2.3. Microbiological Studies

Nasopharyngeal exudates were transported in tubes with a virus preservation medium, both with and without an inactivator. The processing of samples was performed by the RT-PCR. Briefly, the viral genetic material was previously extracted using strips with magnetic balls that selectively bind viral RNA, followed by subsequent washes for isolation. Two different extraction kits were used: the Chemagic Viral DNA/RNA 300 Kit H96 in the Chemagic 360 extractor (PerkinElmer^®^, Lübeck, Germany) and the Versant Simple Preparation 1.0 in the Versant kPCR Molecular System extractor (Siemens^®^ Healthcare Diagnostics, Munich, Germany). For the detection of viral RNA, nucleocapsid (N) and the open reading frame (ORF) 1ab were used as target genes. Positive samples were those with amplification curves at CT ≤ 37 counts; as the amplification of both targets was read in the same channel, only one fluorescence curve was considered. The EURORealTime SARS-CoV-2 (Euroimmun^®^, PerkinElmer^®^, Lübeck, Germany) platform was used for the molecular test. The lowest CT value among all the targets of interest used for amplification was registered only in the positive samples.

### 2.4. Statistical Analysis

In an initial approach, a descriptive analysis of the CT values, cases of SARS-CoV-2 infection, and admissions for COVID-19 was performed. The epidemiological analysis was restricted to cases and hospitalizations in the Community of Madrid, as registered by the Ministry of Health and available in the public domain on its web page [14]. Continuous values were expressed as means and standard deviations. For the univariate analysis, Student’s *t*-test was used to compare means. For the association between proportional values, the X^2^ test was used. In all tests, a *p* < 0.05 was considered the minimum limit of statistical significance.

An analysis of the time series was used to study the association between the curves recording the evolution of the daily cases and hospital admissions (the dependent variables) and the average of daily CT values (independent variable). Laboratory data were available for all seven days of the week.

All series were subsequently deseasonalized by seasonal-trend decomposition, using LOESS (STL) in its robust configuration, with the default parameters treated according to the routine available in the Python statsmodels 0.13.5 library (statsmodels.tsa.seasonal.STL), based on Cleveland et al. [15].

The stationarity of the series was discriminated by applying the standard tests for assessing the presence of a unit root, including the augmented Dickey–Fuller (ADF) test, the Phillip–Perron (PP) test, and the Kwiatkowski–Phillips–Schmidt–Shin (KPSS) test, for which the EViews package (version 12.0, IHS Global Inc., Irvine, CA, USA) was used. Due to the presence of structural changes in the study series, the results of the single-root tests were contrasted with those obtained with the Enders and Lee approximation for structural changes, the Fourier ADF test, and the Fourier–Lagrange multiplier (LM) test, for which GAUSS software (version 22.0, Aptech Systems, Inc., Higley, AZ, USA) was used. After the application of these tests, the hypothesis of no single root at a certain level or first difference was proven. The lag length was based on Akaike information criteria.

The relationship between the CT counts with cases of COVID-19 or hospital admissions was analyzed with the deseasonalized series in a first approach by Maki’s cointegration test (GAUSS software, version 22.0, Aptech Systems, Inc., Higley, AZ, USA) for an unknown number of structural changes in a first approach; Granger-type causality was explored later using the variant described by Toda and Yamamoto (GAUSS software, version 22.0, Aptech Systems, Inc., Higley, AZ, USA). The lag length was based on Akaike information criteria with an upper limit fixed at 12 days, according to previous studies. In addition, the results were contrasted with those obtained with the single Fourier frequency causality Test, due to a better adaptation to the series with structural changes [16].

## 3. Results

### 3.1. Descriptive Analysis

During the period of study, a total of 22,906 nasopharyngeal swabs rendered a positive result for the SARS-CoV-2 RT-PCR test. A total of 14,856 (64.8%) swabs came from primary care centers, 871 (3.8%) came from hospital outpatient clinics, and 7179 (31.4%) came from emergency services. During this period, a total of 422,110 cases of COVID-19 were registered, with a maximum peak of 7863 cases per day, a minimum of 245 cases per day, and a mean of 2069.2 (SD: 1612.4) cases per day. In this same period, a total of 36,727 hospital admissions were also recorded, with a maximum of 517 admissions per day, and a minimum of 72 admissions per day. The mean number of daily admissions was 180.0 (76.4). Table 1 shows the distribution of cases, hospital admissions, and CT tests in different groups by age. The number of cases and CT tests by age groups was comparable (17.2% vs. 16.6% in patients younger than 20 years old; 63.8% vs. 61.4% in patients between 20 and 59 years old; and 18.9% vs. 22.0% patients older in 60 years old or more). Among subjects with a positive RT-PCR test, a total of 12,796 (55.9%) patients were women and 10,110 (44.1%) were men. The overall mean age was 42.5 (SD: 21.7) years (42.6 (SD: 22.3) years for men and 42.4 (SD: 21.3) years for women).

The mean CT value for the entire number of positive RT-PCR tests was 27.4 (SD: 2.1) cycles. The percentage of the analyzed samples that presented a CT below 20 counts was 22.2%, 31.4% presented between 20 and 30 cycles, and 46.4% presented cycles greater than 30.

### 3.2. Analysis of Temporal Series

During the study period, temporal variations were observed for both the mean number of daily cases and hospitalizations for COVID-19, as well as for the mean CT values in the positive RT-PCR tests performed on the same day. Figure 1 shows the time series trend components of this period for hospital admissions, cases, and CT values.

A hypothetical temporal relationship between CT values with cases and daily inpatient ward admissions was explored by a Granger causality test using the Toda and Yamamoto variant. Previously, the stationarity of the series was discriminated as described (Table 2) and the alternative hypothesis of no cointegration was rejected using the Maki cointegration test with significance less than 1% (−14.706; −8.004(1%), −7.414(5%), −7.110(10%), for regime shift and trend). The lags were automatically selected following the Akaike information criterion.

The results for the direction of causality obtained from the causality tests on the selected lags and their statistical significance are shown in Table 3.

As can be seen in the table, there is evidence of Granger causality in the expected direction. This means that, in a theoretical model, the value of CT could improve the prediction of the number of COVID-19 cases and admissions, at least in the selected lags.

## 4. Discussion

Using a temporal series analysis, we showed that the monitoring of CT values among RT-PCR-positive nasopharyngeal samples in a population is temporally correlated with the evolution of cases and hospital admissions caused by the SARS-CoV-2 infection. Therefore, for epidemiological purposes, the reduction in the average CT count may be predictive of a greater number of subsequent cases and hospital admissions of COVID-19 patients.

With limitations, low CT counts indicate a greater SARS-CoV-2 load in the sample [17,18,19,20]. Taking a pathogenic approach, low CT values indicate that diagnosed patients are most likely in the first week of infection [21], when viral loads achieve peak values. Moreover, low CT values are associated with greater chances of detecting viable SARS-CoV-2 in a culture, while the opposite was true for samples with high CT values [7,22]. Lower median CT values were observed in symptomatic cases [23].

A good number of previous studies have shown that CT values, either taken at a given time or analyzing its temporal evolution, may help predict the posterior occurrence of new incident cases or hospitalizations caused by the SARS-CoV-2 infection [11,17,24,25,26,27,28,29,30,31]. To our knowledge, this is the first study to investigate the utility of measuring CT values to predict the evolution of the pandemic using the Granger causality approach, i.e., an analysis of temporal series. Most investigators have chosen a linear regression analysis to prove an association between CT counts and incident episodes of infection. Since we are dealing with the values in the framework of a time series, this approximation, although useful, must be confirmed with other analysis models because of the risk of offering spurious correlations. We believe that it was therefore necessary to complete the evidence linking CT values with the evolution of the pandemic with this type of approximation.

In epidemiological studies, the lag between the CT values and the incidence of cases was approximately 2 weeks [26], while this delay was extended to a full month between the CT values and the number of hospital admissions in another study [30]. A similar delay was observed when cases or hospitalizations were predicted using a SARS-CoV-2 RNA concentration in wastewater [32,33]. According to our analysis based on temporal series, this lag was from 6 to 10 days. The Granger causality test is very sensitive to the lags chosen and the causality may vary depending on the selected lag. Therefore, in our case, we chose a usual selection criterion on the lag length as the Akaike selection criterion with a 12-day upper limit. Having a tool that may herald a rise in symptomatic cases approximately one week in advance may be of great use in the context of SARS-CoV-2 infections. Firstly, this information may recommend the intensification of preventive measures; also, medical centers may bolster personnel and materials at hot spots, such as primary care facilities, emergency services, hospital wards, or intensive care units. Whether population variations in CT values may also be useful in monitoring other epidemic respiratory viruses, such as influenza, warrants study.

The period studied was mostly restricted to the third pandemic wave, when the B.1.177 and B.1.1.7 (Alpha) variants were the most prevalent in Madrid. The SARS-CoV-2 Delta variant, which emerged later in the pandemic, has been shown to have a shorter proliferation phase of infection, peaking sooner, and achieving greater RNA concentrations than these prior variants. The currently circulating Omicron variant has greater transmissibility than any other variant [34,35]. Mutations located at the N-terminal [36] and receptor-biding domains [37] (NTB and RBDs) cause significant evasion of Omicron from neutralizing antibodies, while the hypermutated RBD has enhanced affinity for the angiotensin converting enzyme 2 (ACE2) receptor [38,39]. Several studies have shown that different Omicron sublineages can infect previously infected COVID-19 patients [40,41] and can cause vaccine scape infections [42,43,44]. Booster doses of the mRNA vaccines increase the neutralizing activity against Omicron [45].

Vaccination status, which was that of “no vaccination” in most instances during the period of study, can also change infection dynamics by significantly reducing the clearance phase of infections and by decreasing the total number of days where it is possible to detect the virus [46,47]. The duration of infectivity seems to be comparable among vaccinated and unvaccinated people and between Delta and Omicron variants, despite differences in the immune response or transmissibility, respectively [48]. Still, it is not clear how genetic differences among variants, or the protective effects of vaccination in the population, may affect the ability of the CT count to estimate the dynamics of the pandemic.

The most important limitations of the analysis lie in the behavior of the series. The COVID-19 case series are often high-frequency series, with a high frequency of reporting on weekends or holidays, and with structural changes. These circumstances make it difficult to approach through the study of time series. However, it is known that there are limitations to linear regression or Pearson correlation in this type of data, with the risk of obtaining biased results based on spurious correlations. For this reason, our study applies time series analysis to find this biologically plausible association between the infectivity of the SARS-CoV-2 infection cases, estimated by the CT values, and the number and severity of COVID-19 cases. It is important to remark that Granger causality should not be understood as causality in the epidemiological sense; rather, it indicates a temporal relationship between the past values of one variable and the current values of another. In other words, it suggests the usefulness of one variable in predicting another.

It should be noted that, in our study, two different extraction kits were used, which may have caused different efficiencies in RNA extraction that may have affected the CT values. However, as a single RT-RNA assay was used, the variability of the reported CT values may have been reduced.

The real-time tracking of a pandemic’s local trajectory is essential for resource allocation. Concerns over the potential for local surges in COVID-19 cases prompted many hospitals to preemptively suspend elective procedures in order to strengthen medical staff. Although, as already stated, the utility of CT values is questioned at the individual level, our analysis proves that using this parameter to estimate a rising or abating number of subjects with a high SARS-CoV-2 concentration in the upper respiratory tract may predict the short-term evolution of COVID-19 cases or hospitalizations.

## Figures and Tables

**Figure 1 viruses-15-00421-f001:**
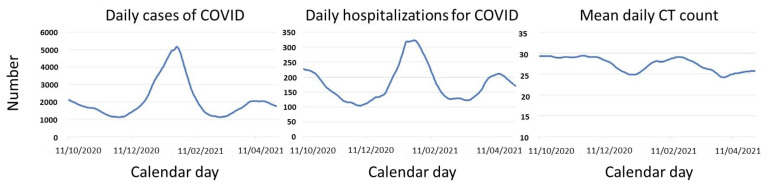
Temporal series analysis for daily cases and hospitalizations by COVID-19 and mean CT counts. CT, cycle threshold in RT-PCR-positive patients for SARS-CoV-2 in nasopharyngeal swabs.

**Table 1 viruses-15-00421-t001:** Characteristics of cases and hospitalizations by COVID-19 and CT counts in positive RT-PCR samples.

	Cases	Admissions	Cycle Threshold	
	N (%)	N (% of Cases)	N (%)	Mean Count (SD)
Total samples	422,110	36,727 (8.7)	22,906	27.4 (2.1)
By age:				
Less than 20 years old	72,582 (17.2)	495 (0.7)	3794 (16.6)	26.7 (3.6)
20 to 59 years old	269,849 (63.8)	12,997 (4.8)	14,060 (61.4)	27.5 (2.5)
60 years old or more	79,640 (18.9)	23,235 (29.2)	5052 (22.0)	27.4 (2.7)
Missing data	39 (0.1)	0 (0)		

**Table 2 viruses-15-00421-t002:** Significance of the null hypothesis rejection of different unit root tests in time series at different levels and first differences.

	Type of Test									
Variables	ADF *		PP *		KPSS *		Fourier LM		Fourier ADF	
	*d*(*0*)	*d*(*1*)	*d*(*0*)	*d*(*1*)	*d*(*0*)	*d*(*1*)	*d*(*0*)	*d*(*1*)	*d*(*0*)	*d*(*1*)
CT counts	NS	1%	1%	1%	NS	NS	NS	10%	NS	1%
Cases of COVID-19	NS	1%	1%	1%	5%	NS	NS	10%	NS	1%
Hospital admissions	NS	10%	NS	1%	10%	NS	NS	5%	NS	5%

(*) With constant and trend. CT, cycle threshold in RT-PCR-positive patients for SARS-CoV-2 in nasopharyngeal swabs; ADF, augmented Dickey–Fuller; PP, Phillip–Perron; KPSS, Kwiatkowski–Phillips–Schmidt–Shin; LM, Lagrange multiplier; NS, not significant.

**Table 3 viruses-15-00421-t003:** Temporal series analysis to associate the evolution of CT counts and cases and the number of COVID-19 hospitalizations.

Type of Temporal Series Test	Direction	Asymmetric *p*-Value	Lag (Days)	Frequency
Toda and Yamamoto Granger causality test	CT → Admissions	0.038	6	0
Single Fourier frequency Toda and Yamamoto causality test	CT → Admissions	0.039	2	2
Toda and Yamamoto Granger causality test	CT → Cases	<0.001	10	0
Single Fourier frequency Toda and Yamamoto causality test	CT → Cases	0.011	9	2

CT, cycle threshold.

## Data Availability

The data presented in this study are available on request from the corresponding author. The data are not publicly available due to ongoing analyses.

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
