# Peer review of "Temporal Series Analysis of Population Cycle Threshold Counts as a Predictor of Surge in Cases and Hospitalizations during the SARS-CoV-2 Pandemic"

_viruses, 2023, doi:10.3390/v15020421_

Round 1

Reviewer 1 Report

Overall, this is a well designed and well written article addressing an important and refreshing angle in the COVID literature: the use of population-level Ct values to help plan for surges in cases and hospitalizations. The authors use statistical tools meant for temporal analyses, Granger-type causality, in a large set of RT-PCR results from the third pandemic wave in Madrid. Their introduction, methods and discussion are thorough. In particular, I value their description of the limitations of Ct values as surrogates for viral loads, their specific study population, and use of high-frequency series with their statistical tool. This work will help advance the conversation about pandemic planning.

Please address the following:

1. Table 1 should be in the results section, not methods, probably between current tables 2 and 3.

2. Description for table 2 mentions sex and healthcare setting, which are not actually included in table 2

3. It would be interesting for the authors to add a paragraph to the discussion about how they would implement this tool in a setting where they can prospectively receive Ct values given the 6-10 day lag

Minor typos, style, etc.

- Please avoid the word 'prove'

- Typos in lines: 82, 264

- 95: should be outpatient clinics

- ? First sentence in 2.4

-139 and 143 should say which, not what

- Legends are split before and after figures

- 220-221 awkward grammar

Author Response

                                                                       Madrid, January 23, 2023

Dear reviewer,

please find attached the revised version of the paper entitled “Temporal Series Analysis of Population Cycle Threshold Counts as a Predictor of Surge of Cases and Hospitalizations During the SARS-CoV-2 Pandemic” intended for publication in the Special Issue of Viruses "State-of-the-Art SARS-CoV-2 Research in Spain". We have incorporated all your corrections and suggestions in the manuscript.

Following there is a point-by-point response to your kind comments:

  1. Table 1 should be in the results section, not methods, probably between current tables 2 and 3. This new allocation has been applied
  2. Description for table 2 mentions sex and healthcare setting, which are not actually included in table 2. This mistake has been corrected.
  3. It would be interesting for the authors to add a paragraph to the discussion about how they would implement this tool in a setting where they can prospectively receive Ct values given the 6-10 day lag. A comment has been included.

Minor typos, style, etc.

- Please avoid the word 'prove'. We have substituted by “analyze.”

- Typos in lines: 82, 264. Corrected

- 95: should be outpatient clinics. Corrected

- ? First sentence in 2.4. The sentence has been rephased.

- 139 and 143 should say which, not what. Corrected

- Legends are split before and after figures. Corrected

- 220-221 awkward grammar. Corrected

We wish to thank you for your time and welcome comments that have improved the clarity and quality of the manuscript,

Pablo Barreiro MD, PhD

Reviewer 2 Report

This article by Cava et al analyses population cycle threshold counts as a predictor of COVID-19 activity. The article warrants publication but needs some minor changes. 

‘COVID’ should be referred to as ‘COVID-19’ throughout. 

Line 45 should read ‘globally more than 650 million confirmed cases’ (remove ‘of’). 

Line 52 ‘the shorter CT’ should read ‘the lower CT’. 

Lines 57 and 63 mention ‘ARN’ – this should be ‘RNA’. 

Line 95 ‘outclinics’ should read ‘outpatient clinics’. 

In section 2.3 Microbiological Studies, two different extraction kits were used. This should be noted in the discussion, as different extraction kits may have different efficiencies in RNA extraction meaning that there may be effects on the CT values. It appears that a single NAT assay was used, which reduces the variability of reported CT values. 

Line 205 states ‘in a population is temporarily correlated’ should read ‘in a population is temporally correlated. 

Line 244 ‘can infect convalescent COVID-19 patients’ should read ‘can infect previously infected COVID-19 patients’. Line 262 ‘estimated by TC values’ should read ‘estimated by CT values’. 

Author Response

Madrid, January 23, 2023

Dear reviewer,

please find attached the revised version of the paper entitled “Temporal Series Analysis of Population Cycle Threshold Counts as a Predictor of Surge of Cases and Hospitalizations During the SARS-CoV-2 Pandemic” intended for publication in the Special Issue of Viruses "State-of-the-Art SARS-CoV-2 Research in Spain". We have incorporated all your corrections and suggestions in the manuscript.

Following there is a point-by-point response to your kind comments:

‘COVID’ should be referred to as ‘COVID-19’ throughout. This change has been introduced.

Line 45 should read ‘globally more than 650 million confirmed cases’ (remove ‘of’). Corrected.

Line 52 ‘the shorter CT’ should read ‘the lower CT’. Corrected.

Lines 57 and 63 mention ‘ARN’ – this should be ‘RNA’. Corrected.

Line 95 ‘outclinics’ should read ‘outpatient clinics’. Corrected

In section 2.3 Microbiological Studies, two different extraction kits were used. This should be noted in the discussion, as different extraction kits may have different efficiencies in RNA extraction meaning that there may be effects on the CT values. It appears that a single NAT assay was used, which reduces the variability of reported CT values. This comment has been introduced.

Line 205 states ‘in a population is temporarily correlated’ should read ‘in a population is temporally correlated. Corrected.

Line 244 ‘can infect convalescent COVID-19 patients’ should read ‘can infect previously infected COVID-19 patients’. Corrected.

Line 262 ‘estimated by TC values’ should read ‘estimated by CT values’. Corrected.

We wish to thank you for your time and welcome comments that have improved the clarity and quality of the manuscript,

Pablo Barreiro, MD, PhD